# Neuropsychological Evidence Underlying Counterclockwise Bias in Running: Electroencephalography and Functional Magnetic Resonance Imaging Studies of Motor Imagery

**DOI:** 10.3390/bs13020173

**Published:** 2023-02-15

**Authors:** Teri Kim, Jingu Kim, Sechang Kwon

**Affiliations:** 1Institute of Sports Science, Kyungpook National University, 80 Daehak-ro, Buk-gu, Daegu 41566, Republic of Korea; 2Department of Physical Education, Kyungpook National University, 80 Daehak-ro, Buk-gu, Daegu 41566, Republic of Korea; 3Department of Humanities & Arts, Korea Science Academy of KAIST, 105-47, Baegyanggwanmun-ro, Busanjin-gu, Busan 47162, Republic of Korea

**Keywords:** turning bias, directional bias, EEG, ERP, fMRI, motor imagery

## Abstract

We aimed to answer the question “why do people run the track counterclockwise (CCW)?” by investigating the neurophysiological differences in clockwise (CW) versus CCW direction using motor imagery. Three experiments were conducted with healthy adults. Electroencephalography (EEG) was used to examine hemispheric asymmetries in the prefrontal, frontal, and central regions during CW and CCW running imagery (*n* = 40). We also evaluated event-related potential (ERP) N200 and P300 amplitudes and latencies (*n* = 66) and conducted another experiment using functional magnetic resonance imaging (fMRI) (*n* = 30). EEG data indicated greater left frontal cortical activation during CCW imagery, whereas right frontal activation was more dominant during CW imagery. The prefrontal and central asymmetries demonstrated greater left prefrontal activation during both CW and CCW imagery, with CCW rotation exhibiting higher, though statistically insignificant, asymmetry scores than CW rotation. As a result of the fMRI experiment, greater activation was found during CW than during CCW running imagery in the brain regions of the left insula, Brodmann area 18, right caudate nucleus, left dorsolateral prefrontal cortex, left superior parietal cortex, and supplementary motor area. In the ERP experiment, no significant differences were found depending on direction. These findings suggest that CCW rotation might be associated with the motivational approach system, behavioral activation, or positive affect. However, CW rotation reflects withdrawal motivation, behavioral inhibition, or negative affect. Furthermore, CW rotation is understood to be associated with neural inefficiency, increased task difficulty, or unfamiliarity.

## 1. Introduction

If asked to run on a curved track, which direction would people choose to run? According to research, most people prefer the counterclockwise (CCW) direction when running on a track [1]. Similarly, Golomer et al. [2] reported rightward turning bias in trained classical dancers during spontaneous rotations. Interestingly, for track events, such as sprinting, hurdles, cycling, and speed skating, the prescribed direction of movement is predominantly CCW. This is applicable to events that involved minimal physical movements, such as horseracing and motor sports, as well. However, there is a lack of scientific evidence to explain the preference for CCW over clockwise (CW) rotation, and why the majority of sporting events have made CCW rotation as the standard for movement.

Most academic research on this question has shown behavioral analyses to investigate directional bias in humans and animals [1,3,4,5,6]. To compare directional preference of participants in the United States and England, Scharine and McBeath [7] conducted an experiment using a simple “T-maze” task, and they suggested that walking direction preference is attributable to both learned driving patterns and genetic handedness. In another study that compared the turning tendency in able-bodied and amputee participants, able-bodied participants who were right-hand dominant exhibited leftward (CCW) turning preference, whereas the turning preference of the amputee sample was not associated with handedness, footedness, or side of amputation, suggesting that biomechanical asymmetries may affect turning bias [8]. Previous studies on turning bias have reported conflicting results and have speculated that directional preference might originate from interactions of internal, external, biomechanical, and anthropometric asymmetries.

Tavakkoli and Jose [9] attempted to answer the question of why athletes run around the track CCW from various perspectives. For example, they adopted embryological viewpoints based on the history of ancient Rome and Greece and presented biomechanical, physiological, and evolutionary reasons, which depended largely on literature-based assumptions and hypotheses rather than empirically verified evidence. They also pointed to natural causes, explaining that everything in nature tends toward CCW motion such as the molecular structure of amino acids and the shape of seashells [9]. Others argued that running CCW would give athletes a slight advantage in terms of faster time, affected by the Earth’s rotation [10].

However, basic observations reveal that human behavior occurs in a direction that is easy, convenient, or emotion-driven. For instance, when researchers asked people to express emotion while posing for a family portrait, they tended to turn rightward to present the left side of the face, compared to posing as a scientist, with restrained emotions, participants tended to turn their head leftward to present the right side [11]. Other researchers have argued that people naturally tend to turn in the direction that they feel most comfortable and involve the least amount of energy, depending on positional constraints. [12]. These findings illustrate that psychological factors influence the choice of turning direction. In this regard, the CCW turning bias is not simply attributable to convention and habit but may be associated with neurophysiological factors. However, few studies have provided neurophysiological data on the same.

Therefore, we aimed to answer the question, “why do people prefer to run in CCW direction around the track?” through neurophysiological investigation. Electroencephalography (EEG) and functional magnetic resonance imaging (fMRI) were adopted, which offer high temporal and spatial resolutions, respectively, to examine the neurophysiological mechanisms underlying the turning bias in humans. In line with the purpose and methods of this study, widely used motor imagery (MI) paradigms were used in neuroscience research in movement-constrained environments [13,14]. Without any overt movements, MI shares cortical representations with motor execution by mentally simulating an action [15].

We aimed to elucidate the neural mechanisms underlying turning bias by investigating neurophysiological differences during MI of running the track in CW versus CCW directions using EEG and fMRI measurements.

## 2. Methods

### 2.1. Participants

A healthy adult cluster of 40 adults (21.65 ± 1.35 years, women = 21) participated in the EEG study, 66 adults (21.71 ± 1.42 years, women = 30) participated in the event-related potential (ERP) study, and another group of 30 adults (21.73 ± 1.52 years, women = 17) participated in the fMRI study. Participants who were not eligible for EEG or fMRI measurements and had poor mental imagery abilities were prescreened and excluded based on responses to the eligibility questionnaire and questionnaire upon motor imagery (QMI). The EEG experiment was conducted based on a between-subject design, where the participants were randomly assigned to either CW or CCW groups, depending on the direction of movement imagery they were required to follow. The fMRI experiment employed a within-subject design, which required all participants to perform movement imagery in both the CW and CCW conditions. All participants were self-declared to be right-handed and had normal or corrected-to-normal vision. None of the participants had any history of neurological disorders or brain diseases or contraindications to undergo EEG or fMRI testing. After eliminating data with artifacts or poor image resolution, the EEG data of 36 participants (17 women) and fMRI data from 26 participants (13 women) were used for the final analysis. Written informed consent was obtained from all participants prior to the study, and the study protocol was approved by the institutional review board of Kyungpook National University (No. 2017-0074).

### 2.2. Instruments and Paradigm

#### 2.2.1. Questionnaire upon Mental Imagery (QMI)

Prior to the EEG and fMRI experiments, participants completed the QMI developed by Sheehan [16] and adapted for Korean audiences by Park and Park [17] for the assessment of mental imagery ability. The QMI is a 35-item measure consisting of seven subscales, with five items each for visual, kinesthetic, auditory, olfactory, gustatory, tactile, and cutaneous sensory modalities. Based on a 7-point vividness rating scale (0: no imagery, 7: imagery as vivid as real), the total score ranges between 35 and 245, with a lower score indicating greater imagery vividness. Cronbach’s alpha value of the QMI in our EEG experiment was 0.96 with no significant difference between the CW (mean = 171.0) and CCW groups (mean = 176.8) (*t* = −0.67, df = 18, *p* = 0.91). Cronbach’s alpha value of the QMI in the fMRI experiment was 0.96.

#### 2.2.2. Experimental Paradigms for the EEG and ERP Studies

To measure asymmetrical hemispheric activation using EEG, an MI task with a first-person view was designed, in which participants were asked to imagine running on the curved section of the track on the screen for 3 min (Figure 1a). 

For the ERP study, the MI paradigm was modified from a previous study [18], which was developed based on the traditional arrow paradigm, widely used in MI studies [19,20] (Figure 1b). A trial (10 s) consisted of “Ready” sign (2 s), followed by a fixation (2 s), an MI cue (6 s), and a blank screen (2 s). All study participants performed four blocks of 50 trials, totaling to 200 trials. When the MI cue (i.e., an image of the running track curved to either the CW or CCW direction) appeared after fixation, the participants were instructed to imagine running the track in the cued direction for 6 s and then rest during the presence of the blank screen. Sufficient rest time was provided between the blocks.

#### 2.2.3. Experimental Paradigm for the fMRI Study

For the fMRI study, the experimental paradigm was adopted from the traditional arrow paradigm used in previous MI studies [18,21], and was modified according to the purpose of the study (Figure 1c). The experimental paradigm comprised 60 trials, where 15 trials of four conditions (i.e., CW MI, CCW MI, neutral imagery, and fixation) were presented in a randomized order for a total of 480 s, each lasting 8 s. In the MI conditions, three connected track images of 2 s each were presented consecutively for a total of 6 s, followed by a question for two s asking about subjective feelings during the imagery. Participants were instructed to respond to “How did you feel about the running direction of the imagery?” by choosing one option (1 = very uncomfortable, 2 = uncomfortable, 3 = comfortable, 4 = very comfortable) and pressing the corresponding button on the response pad. An experimental task was developed using E-prime 2.0.

### 2.3. EEG Acquisition and Analysis

A Biopac MP150 system (Biopac System Inc., Santa Barbara, CA, USA) was used for EEG data acquisition. When the participants arrived at the laboratory, they were explained the purpose and procedure of the study, including instructions about the experimental paradigm. They proceeded to sign an informed consent form and remove any metallic materials and electronic devices from their bodies. A Lycra cap (Electro-cap: EM1) housed with Ag/AgCl electrodes positioned according to the international 10–20 system [22] was fitted to the participant’s head, and conductive gel was injected into the electrodes at the following regions of interest: Fp1 (left prefrontal), Fp2 (right prefrontal), F3 (left frontal), Fz (mid frontal), F4 (right frontal), C3 (left central), Cz (mid central), C4 (right central), P3 (left parietal), Pz (mid parietal), and P4 (right parietal). A reference electrode was attached to both earlobes with Fpz serving as the ground electrode.

An electrode for electrooculography (EOG) recording was attached adjacent to the left eye, and the impedance levels of all channels were maintained below 5 kΩ throughout the EEG recording using a grass impedance meter (EZM5, Astro-Med Inc., West Warwick, RI, USA). Continuous EEG and EOG signals were recorded at a sampling rate of 1000 Hz. The experimental stimuli were presented on a monitor (1920 × 1080 cm) placed 1.3 m away from the seated participant. Each participant’s baseline EEG was recorded for 3 min each, with eyes closed and with eyes open in a relaxed state. The participants then performed the experimental task for EEG (Figure 1a) for 180 s, followed by the experimental task for ERP (Figure 1b) consisting of four blocks of 50 trials. Real-time EEG data were collected during task performance. 

The acquired EEG signals were 1–35 Hz band-pass filtered and digitized at a 256 Hz sampling rate by channels using the Acqknowlege 4.2 (Biopac system Inc., Goleta, CA, USA) software program. A finite impulse response band-pass filter was used to filter the digitized signals in the 6–20 Hz band. Signals with amplitudes exceeding ± 100 μV or those contaminated by artifacts were inspected and excluded. The processed data were divided into 1 s window chunks, and then fast-Fourier transform analysis was performed in the alpha (8–13 Hz) frequency band. Through this process, the digitized EEG alpha power was obtained, which was log-transformed to calculate the log power density for each electrode site. The EEG asymmetry values were obtained by subtracting the left-sided alpha power from the right-sided alpha power (log right minus log left alpha power) [23]. As cortical alpha power is inversely related to cortical activity, positive (+) scores (higher alpha power in the right hemisphere) correspond to relatively greater left hemisphere activation, while negative (−) scores (higher alpha power in the left hemisphere) indicate relatively greater right hemisphere activation [24].

To analyze the ERP measures, EEG data were collected from the Fz, Cz, and Pz regions. Baseline corrections were conducted using the mean 250 ms prestimulus period, and the data were digitized at a sampling rate of 1000 per channel and band-pass-filtered (IIR filter) in the 0.1–30 Hz band. Signals containing artifacts or EOG were removed. To extract a stimulus-locked epoch, data were segmented into epochs of 2000 ms. Among the ERP components, the peak amplitude and corresponding latency of N200 and P300 were extracted. N200 is a negative potential that appears at a latency of 200–300 ms after stimulus presentation, and P300 is a positive potential with a latency of approximately 300 ms after stimulus presentation. All the EEG data were processed and analyzed using the Acqknowlege 4.2 and Matlab R2019a software.

### 2.4. fMRI Acquisition and Analysis

A 3T GE Unit (Signa Signa EXCITE HD; GE Healthcare, Piscataway, NJ, USA) was used to acquire the fMRI data. After being informed of the purpose of the study, experimental procedure, and precautions, the participants provided written informed consent. They were instructed to perform practice trials until they completely understood the experimental task. The participants were instructed to change their clothes, rest, and then enter the scanning room to lie inside the fMRI chamber with their body and head fixed to minimize motion. During the entire scanning process, the participants were required to avoid psychological activity as much as possible. The experimental tasks were delivered through a screen installed in front of the participant’s eyes, and the scanning time was 6 min and 8 s. 

BOLD functional images were acquired (EPI, TR = 2000 ms, TE = 30 ms, matrix = 64 × 64, Thickness = 3.0 mm, FOV = 220 mm, no gap). A 3D T1-weighted anatomical scan was obtained for structural reference. For data analyses, MATLAB R2019a (Mathworks Inc., Natick, MA, USA) and SPM12 (SPM; Wellcome Department of Imaging Neuroscience, London, UK) programs was used. Slice-timing correction and realignment for temporal and spatial corrections, respectively, followed by spatial normalization of the structural image to a standard template (Montreal Neurological Institute, MNI), was applied. Then, spatial smoothing was performed using an isotropic Gaussian filter kernel with full width at half maximum. 

By applying a general linear model, the functional timeline of data was regressed to the repeated task effects and hemodynamic response function (HRF). Low-frequency noise was eliminated using a standard high-pass filter with a 128 s cut-off, and the effect of the HRF caused by repeated presentation of the task was eliminated with a low-pass filter of the frequency suggested by SPM12. A general linear model was used to analyze the activation areas of the brain and calculate the individual activation estimates.

### 2.5. Statistical Analysis

To determine differences in hemispheric asymmetry depending on the direction (CW vs. CCW), independent sample t-tests was performed on the asymmetry scores of the two groups, derived by the formula (log R–log L alpha power) for each of the regions of interest: prefrontal (Fp2-Fp1), frontal (F4-F3), and central (C4-C3). To determine differences in the ERP components depending on the direction, 2 (group: CW vs. CCW) × 3 (regions: Fz, Cz, Pz) × 4 (block) repeated-measures ANOVAs were performed separately on N200 and P300 peak amplitudes and latencies. 

To examine differences in brain activation depending on MI direction, the collected fMRI data were analyzed for (1) CW vs. neutral, (2) CCW vs. neutral, (3) CW vs. CCW, and (4) CCW vs. CW. For within-group analyses of fMRI data, contrast images from the analysis of individual participants in different directional conditions were analyzed using a one-sample *t*-test, thereby generating a random-effects model, allowing inference to the general population. 

All statistical analyses were performed using SPSS 22.0, and the statistical significance level was set at 0.05. 

## 3. Results

### 3.1. EEG Hemispheric Asymmetry

Differences in hemispheric asymmetry scores according to the direction of running imagery in the prefrontal (Fp2-Fp1), frontal (F4-F3), and central (C4-C3) regions are presented in Table 1. The asymmetry scores of the prefrontal region indicated no significant differences as a function of direction (*t* = −0.973, df = 34, *p* = 0.338). However, both CW and CCW MI groups exhibited higher activation in the left prefrontal region (Fp1) than in the right prefrontal region (Fp2).

In the analysis of the frontal asymmetry scores, a significant difference was found as a function of direction (*t* = 6.090, df = 34, *p* < 0.001). The CW direction was associated with higher right frontal (F4) activation relative to the left frontal region (F3), whereas the CCW direction demonstrated left-hemispheric dominance (Figure 2). 

The central hemispheric asymmetry (C4-C3) of CCW and CW MI was not significantly different (*t* = 1.776, df = 34, *p* = 0.088). However, both CCW and CW MI directions were associated with greater left-central activation than right-central activation (Table 1).

### 3.2. ERPs

In the analysis of the N200 amplitudes, no significant main effect of group was observed (i.e., MI direction) (F [1,64] = 0.704, *p* = 0.405, np2 = 0.011). The interaction effects of group × region × block (F [6,384] = 0.824, *p* = 0.552, ηp2 = 0.013), group × region (F [2,128] = 0.226, *p* = 0.798, ηp2 = 0.004), and group × block (F [3,192] = 0.081, *p* = 0.970, ηp2 = 0.001) were not statistically significant. In the analysis of the N200 latencies, no significant main effect of group was observed (i.e., MI direction) (F [1,64] = 0.429, *p* = 0.515, ηp2 = 0.007). The interaction effects of group × region × block (F [6,384] = 1.114, *p* = 0.353, ηp2 = 0.017), group × region (F [2,128] = 0.670, *p* = 0.514, ηp2 = 0.010), and group × block (F [3,192] = 1.186, *p* = 0.316, ηp2 = 0.018) were not statistically significant. The N200 amplitudes and latencies time-locked to the onset of MI in the CW and CCW directions by region are presented in Table 2. 

In the analysis of the P300 amplitudes, significant main effect of group (i.e., MI direction) was absent (F [1,64] = 0.055, *p* = 0.816, ηp2 = 0.001). The interaction effects of group × region × block (F [6,384] = 0.861, *p* = 0.524, ηp2 = 0.013), group × region (F [2,128] = 0.051, *p* = 0.950, ηp2 = 0.001), and group × block (F [3,192] = 0.261, *p* = 0.854, ηp2 = 0.004) were not statistically significant. 

In the analysis of the P300 latencies, no statistically significant main effects of group (i.e., MI direction) (F [1,64] = 5.784, *p* = 0.05, ηp2 = 0.083), nor significant interactions of group × region × block (F [6,384] = 1.100, *p* = 0.362, ηp2 = 0.017), group × region (F [2,128] = 0.467, *p* = 0.628, ηp2 = 0.007), and group × block (F [3,192] = 0.371, *p* = 0.774, ηp2 = 0.006) were observed. The P300 amplitudes and latencies time-locked to the onset of MI in the CW and CCW directions by region are presented in Table 2.

### 3.3. fMRI

The analysis of fMRI data revealed that the CW direction was associated with greater activation in the left insula (*t* = 4.15, *n*= 26, *p* < 0.05), Brodmann area 18 (BA 18) (*t* = 5.55, *n* = 26, *p* < 0.05), right caudate nucleus (CN) (*t* = 3.49, *n* = 26, *p* < 0.05), left dorsolateral prefrontal cortex (DLPFC) (*t* = 4.54, *n* = 26, *p* < 0.05), left superior parietal cortex (SPC) (*t* = 4.71, *n* = 26, *p* < 0.05), and supplementary motor area (SMA) (*t* = 4.80, *n* = 26, *p* < 0.05) than the CCW direction (Figure 3).

## 4. Discussion and Conclusions

We investigated the neural correlates of CW and CCW rotations during MI using EEG and fMRI to provide neurophysiological evidence for why people tend to run on a track in CCW direction. The EEG data revealed prefrontal and frontal asymmetries depending on the direction. Specifically, greater left frontal cortical activation was observed during CCW imagery, whereas right frontal activation was dominant during CW imagery. Furthermore, greater left prefrontal activation was found during both CW and CCW imageries, with CCW rotation exhibiting a greater left frontal asymmetry than CW rotation.

Frontal EEG alpha asymmetry, the difference index between the left and right frontal activation, reflects motivational approach or avoidance, behavioral activation or inhibition, and positive or negative affective appraisal [25,26,27]. The results of the study illustrate that CCW running imagery, with greater relative left frontal activation, might be associated with the approach motivation system, behavioral activation, or positive affect. On the other hand, CW running imagery, showing greater right frontal activation, was more relevant to the avoidance motivation system, behavioral inhibition, or negative affect. These findings provide neurophysiological evidence supporting previous findings that people feel more familiar and comfortable running CCW than CW, and hence prefer to move in the CCW direction. This interpretation is supported by a previous study that reported left hemisphere activation when processing familiar stimuli and right hemisphere activation when processing unfamiliar stimuli [28].

As the participants were less familiar with CW rotation than with CCW rotation, it was more difficult for them to perform MI in the CW direction. This was also reflected in the fMRI data obtained in the present study. Analyses revealed greater activation during CW than during CCW running imagery in the regions of the left insula, BA 18, right CN, left DLPFC, left SPC, and SMA.

Greater brain activation can be understood in the context of neural inefficiency [29]. In addition, Mizuguchi and Kanosue [30] reported that an increase in task difficulty is associated with a greater intensity of blood oxygenation level-dependent signals and area recruitment. Considering that brain activation may increase in parallel with task difficulty, causing a decline in neural efficiency, the CW MI performed in the present study might have been more difficult than the CCW imagery. As the participants were less familiar with running the track in the CW direction, it may have required more control and focused attention, resulting in the recruitment of broader brain regions with greater neural activation.

The brain areas exhibiting significantly greater activation during CW imagery than during CCW imagery largely overlapped with the areas noted in the study by Thobois et al. [31], where participants performed MI with either the left or right hand. They observed DLPFC, SPC, and SMA activation only during MI with the left (non-dominant) hand, but not for the right hand. In particular, the SMA, a portion of the premotor cortex, is the center of motor planning and execution [32]. SMA has been reported to be the most active area involved in MI, high-level motor control, and movement programming [33,34,35]. Consistent with previous studies, the greater recruitment of multiple areas involved in motor preparation observed in the present fMRI study during CW relative to CCW imagery may indicate relative difficulty in imagining less familiar and less automated movements. Comparatively, MI in the CCW direction might have been easier and, hence, required less cortical activation.

Previous studies have shown that insular activation is associated with awareness of body parts [36], detection of novel stimuli across sensory modalities [37], and mental navigation along memorized routes [38]. Furthermore, BA 18, part of the occipital visual area, plays a role in receiving information from the primary visual cortex (BA 17) and providing input on spatial vectors to the middle temporal and medial superior temporal cortices. BA 18 is of primary importance in visual depth perception and gaze control [39]. Therefore, the results of the present study illustrate that CW running imagery requires more mental effort than CCW running imagery to obtain a sense of the physical condition of the body and to detect and utilize visual information.

The CN, a pair of brain structures located in the basal ganglia, play a key role in processing visual information and controlling movement, and are specifically recruited during visual imagery [40]. Sauvage et al. [41] established recruitment of the right CN during slow movement imagery as an unusual motor task that required voluntary central control and attention to fine-tune the appropriate submovements in the motor sequence based on sensory feedback, similar to the results of the present study. In another study, CN activation increased during the initiation of a turn, which was most pronounced contralateral to the intended direction [42]. This supports the greater right CN activation during left turn (CW) imagery found in our study, confirming that the CN can be recruited during imagery of contraversive body movements for visual information processing and fine motor control.

In addition, the analysis showed greater activation in the DLPFC during CW than during CCW MI. Owing to the involvement of the DLPFC in the evaluation of accumulated information and response selection, it can be safely assumed that greater DLPFC activation during CW imagery in this study is indicative of more effortful monitoring and decision-making processes [43]. In another study, together with the anterior cingulate cortex, the DLPFC showed greater activity for unsuccessful performance than for successful performance of imagery and memory retrieval, which was also associated with low levels of confidence [44]. Therefore, the participants performing CW imagery in this study may have experienced difficulties in concretizing an image by retrieving memory, which demanded greater recruitment of the DLPFC function to perform the task.

Greater activation of the SPC during CW than during CCW running imagery may indicate that more attention and cognitive processing are required to plan and execute CW running. The parietal region, including the SPC, is responsible for higher-level cognitive functions, such as reasoning and problem-solving [45]. This interpretation is supported by the findings of Behrmann, Geng, and Shomstein [46], who observed increased SPC activation when the concentration on task performance increased. In an fMRI study that analyzed brain activation while virtually driving familiar versus unfamiliar routes, more visual attention was required under unfamiliar routes, with greater activation of the parietal region [47,48]. Therefore, SPC activation during CW running imagery in this study might be attributable to learned experiences (i.e., familiarity) of movement directions.

In summary, the EEG alpha asymmetry measures indicate that CCW running imagery induced greater left hemisphere activation, reflecting the motivational approach system, behavioral activation, or positive affect. In contrast, greater right hemisphere activation was observed during CW running imagery, relevant to the motivational withdrawal system, behavioral inhibition, or negative affect. Furthermore, the fMRI measurements illustrated greater activation in the left insula, BA 18, right CN, left DLPFC, left SPC, and SMA during CW than during CCW running imagery. Greater recruitment of multiple areas involved in movement preparation and execution during MI can be viewed as evidence of neural inefficiency, task difficulty, or task unfamiliarity related to CW rotation. Based on these neuroimaging data, this study provides a scientific rationale for why people run a track in the CCW direction.

However, it remains unclear whether these neurophysiological differences between CW and CCW running are products of learning and experience or whether the differences are innate characteristics from the beginning, such that people are naturally induced to move in the CCW direction. Therefore, we encourage future studies to further investigate neural networks related to motor learning and memory and invite younger participants with minimal learning experience in the direction of movement initiation.

## Figures and Tables

**Figure 1 behavsci-13-00173-f001:**
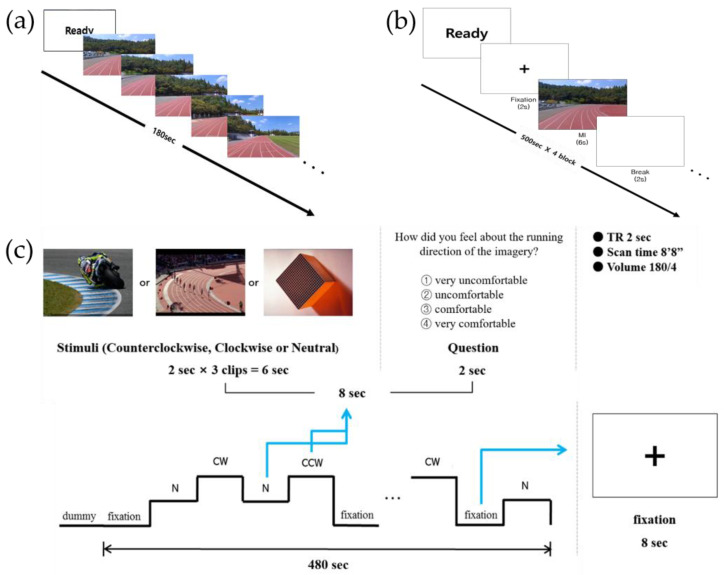
Experimental tasks used in the (**a**) electroencephalography, (**b**) event-related potential, and (**c**) functional magnetic resonance imaging studies.

**Figure 2 behavsci-13-00173-f002:**
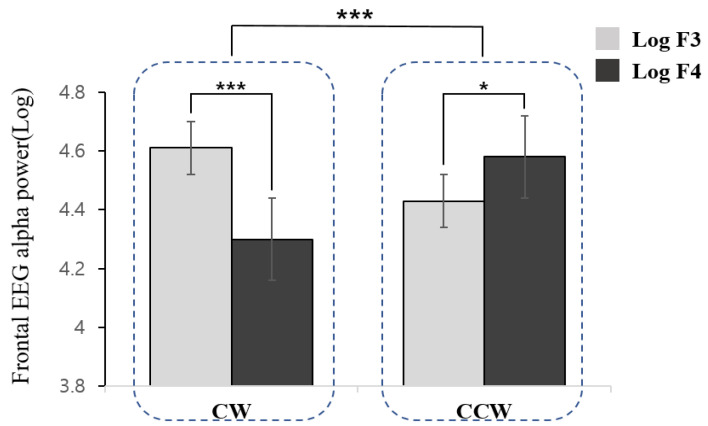
Frontal electroencephalography alpha power as a function of direction of movement imagery (CW: Clockwise, CCW: Counterclockwise) and hemispheres (F3: left frontal, F4: right frontal). *** (*p <* 0.001) and * (*p <* 0.05).

**Figure 3 behavsci-13-00173-f003:**
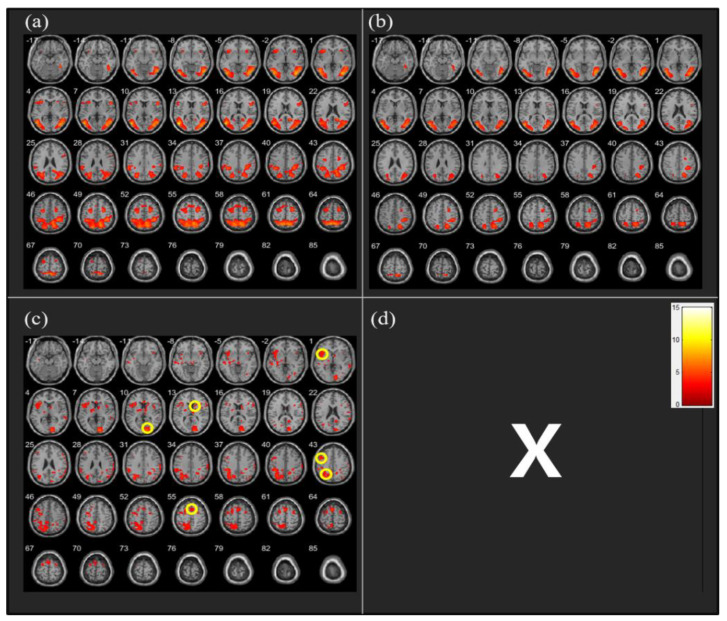
Neural activity during clockwise (CW) vs. counterclockwise (CCW) running imagery (**a**) CW > Neutral: no significant differences, (**b**) CCW > Neutral: no significant differences, (**c**) CW > CCW: significantly different activation observed in the following regions [1] left insula, [10] Brodmann area 18, [13] right caudate nucleus, [43] left dorsolateral prefrontal cortex, left superior parietal cortex, and [55] supplementary motor area, (**d**) CCW > CW: no significant differences.

**Table 1 behavsci-13-00173-t001:** Electroencephalography (EEG) alpha asymmetry scores according to the direction of motor imagery at the prefrontal (Fp2-Fp1), frontal (F4-F3), and central (C4-C3) regions.

Region	EEG Alpha Asymmetry Score			
	Clockwise		Counterclockwise	
	M (SD)	Range	M (SD)	Range
Log Fp1	5.09 (1.97)	1.88–6.62	5.29 (1.53)	−0.64–6.85
Log Fp2	5.20 (1.99)	0.18–6.62	5.32 (1.76)	−1.56–7.00
Log Fp2-Fp1	0.11 (0.19)	−0.18–0.66	0.27 (0.27)	−0.91–0.31
Log F3	4.60 (1.66)	2.14–4.92	4.43 (1.31)	1.40–5.53
Log F4	4.30 (1.60)	2.27–4.68	4.57 (1.25)	1.84–5.85
Log F4-F3	−0.31 (0.20)	−0.74–0.56	0.14 (0.24)	−0.21–0.61
Log C3	3.48 (1.24)	1.64–5.97	3.06 (1.56)	−1.59–5.32
Log C4	3.87 (1.40)	0.41–6.04	4.25 (1.47)	−1.43–5.37
Log C4-C3	0.38 (0.90)	−1.23–1.96	1.20 (1.73)	−3.21–6.14

**Table 2 behavsci-13-00173-t002:** Event-related potential (ERP) N200 and P300 amplitude and latency time-locked to the onset of motor imagery in CW vs. CCW directions.

Components		Event-Related Potential (ERP)					
		Clockwise M (SD) *n* = 33			Counterclockwise M (SD) *n* = 33		
		Fz	Cz	Pz	Fz	Cz	Pz
N200	amplitude (mV)	−22.75	−52.75	−62.42	−30.45	−57.79	−65.74
		(21.20)	(23.56)	(29.21)	(35.10)	(33.65)	(35.10)
	latency (ms)	269.65	277.99	288.65	264.83	278.75	279.32
		(31.93)	(31.89)	(32.01)	(37.94)	(31.89)	(39.83)
P300	amplitude (mV)	35.16	40.95	49.76	38.68	42.31	50.95
		(37.71)	(48.97)	(43.49)	(33.77)	(35.82)	(37.94)
	latency (ms)	424.70	358.96	380.69	475.11	420.87	409.46
		(131.62)	(98.23)	(86.71)	(145.01)	(100.20)	(111.07)

## Data Availability

Not applicable.

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
