# Peer review of "Neuropsychological Evidence Underlying Counterclockwise Bias in Running: Electroencephalography and Functional Magnetic Resonance Imaging Studies of Motor Imagery"

_behavsci, 2023, doi:10.3390/bs13020173_

Round 1
Reviewer 1 Report
The topic of the paper is interesting and fits the scope of the journal. The text is relatively well written and composed. I have only minor comments that I believe that help to improve the paper. Congratulations to the authors for this good work.
You can add my following thoughts in introduction and conclusion:
I think that a possible explanation about the counterclockwise is because everything in nature towards counterclockwise motion. For example, the molecule structure of amino acids and the orbital direction of the earth around the sun (Tavakkoli & Jose, 2013). One other explanation is because of the effects of Earth’s rotation, an athlete running anticlockwise will have resulting in faster time (Brown, 2011).
Minor comments
Lines 102-103: Please report the name of university.
Figure 2: Please explain with any symbol if there are any significant differences between F3 and F4 CW and CCW.
Brown, P. (2011). Why do athletes have to race around the track in an anti-clockwise direction?, Notes & Queries. The Sporting Life. London: guardianNews.://www.guardian.co.uk/notesandqueries/query/0,5753,-1416,00.html.
Tavakkoli, M.H and Jose, T.P. (2013). The reason why do athletes run around the track counter-clockwise. International Educational E-Journal, 2, 4.
Author Response
Dear Reviewer,
Thank you for your kind comments. We feel that the manuscript has improved substantially in many aspects owing to your support. Please refer to each of the responses below for revision details and see the changes within the manuscript highlighted in red.
Reviewer 1
The topic of the paper is interesting and fits the scope of the journal. The text is relatively well written and composed. I have only minor comments that I believe that help to improve the paper. Congratulations to the authors for this good work.
You can add my following thoughts in introduction and conclusion:
I think that a possible explanation about the counterclockwise is because everything in nature towards counterclockwise motion. For example, the molecule structure of amino acids and the orbital direction of the earth around the sun (Tavakkoli & Jose, 2013). One other explanation is because of the effects of Earth’s rotation, an athlete running anticlockwise will have resulting in faster time (Brown, 2011).
Brown, P. (2011). Why do athletes have to race around the track in an anti-clockwise direction?, Notes & Queries. The Sporting Life. London: guardianNews.://www.guardian.co.uk/notesandqueries/query/0,5753,-1416,00.html.
Tavakkoli, M.H and Jose, T.P. (2013). The reason why do athletes run around the track counter-clockwise. International Educational E-Journal, 2, 4.
- Thank you for the suggestion. We have included the following sentences in the Introduction section. Please refer to line 59-63.
“They also pointed to natural causes, explaining that everything in nature tends towards CCW motion such as the molecular structure of amino acids and the shape of seashells [9]. Others argued that running CCW would give athletes a slight advantage in terms of faster time, affected by the Earth’s rotation [10].
Minor comments
Lines 102-103: Please report the name of university.
- The name of the university has been inserted as below.
“Written informed consent was obtained from all participants prior to the study, and the study protocol was approved by the institutional review board of Kyungpook National University (No. 2017-0074).”
Figure 2: Please explain with any symbol if there are any significant differences between F3 and F4 CW and CCW.
- Figure 2 has been revised between line 280 and 281. There were significant differences between F3 and F4 in both CW(p<.001) and CCW(p<.05).
Thank you very much.
..................................................................
(For 2nd revision)
Dear Reviewer,
Thank you once again for your kind comments. When I checked the second review results, I could not find your comments. In order to make our draft better, we additionally checked and fixed the English grammar and spelling. If your second comment did not reach us for some reason, please send it again, and we will gladly revise it.
Thank you very much for taking the time out of your busy schedule.

Reviewer 2 Report
This is an interesting paper “why do people run the track counterclockwise 13 (CCW)?” by investigating the neurophysiological differences in clockwise (CW) versus CCW direction using motor imagery. The concept is novel due to the current limit understanding of this common phenomenon. However, there are several issues regarding to the organization of the paper.
1. The author claims that “CCW rotation might be associated with the motivational approach system, behavioral activation, or positive affect. However, CW rotation reflects withdrawal motivation, behavioral inhibition, or negative affect”. The reason is not very clear in the article. Is it more possible due to “unfamiliarity”? People feel extremely uncomfortable when running CW which shows in EEG and fMRI. What if the author adds another group of 40 people, let them run CW for one week and get them used to it. And then compare the CW and CCW signal.
2. The references are expected to be updated. Only 5/47 references were published after 2017. For a research article of 2023, most of the references are expected to be from after 2017 and summarize more recent research progression.
Author Response
Dear Reviewer,
Thank you for your kind comments. We feel that the manuscript has improved substantially in many aspects owing to your support. Please refer to each of the responses below for revision details and see the changes within the manuscript highlighted in red.
Reviewer 2
This is an interesting paper “why do people run the track counterclockwise 13 (CCW)?” by investigating the neurophysiological differences in clockwise (CW) versus CCW direction using motor imagery. The concept is novel due to the current limit understanding of this common phenomenon. However, there are several issues regarding to the organization of the paper.
- The author claims that “CCW rotation might be associated with the motivational approach system, behavioral activation, or positive affect. However, CW rotation reflects withdrawal motivation, behavioral inhibition, or negative affect”. The reason is not very clear in the article. Is it more possible due to “unfamiliarity”? People feel extremely uncomfortable when running CW which shows in EEG and fMRI. What if the author adds another group of 40 people, let them run CW for one week and get them used to it. And then compare the CW and CCW signal.
- Thank you for your insightful advice. We agree with your point that the reason is not very clear and take is as a limitation of this study. However, as you can see in the discussion, we provided several possible interpretations of the results focusing on unfamiliarity along with task difficulty and neural inefficiency, which we think may affect the motivational system. We also mentioned in the discussion “Therefore, SPC activation during CW running imagery in this study might be attributable to learned experiences (i.e., familiarity) of movement directions.” Since this is a preliminary study, we plan to conduct a follow-up study, where as you suggested, another group will be designed to determine whether the differences between CCW vs. CW shown in this study are due to a learned effect.
- The references are expected to be updated. Only 5/47 references were published after 2017. For a research article of 2023, most of the references are expected to be from after 2017 and summarize more recent research progression.
- Thank you for the suggestion. We have updated the references, however, some essential references, were published before 2017, used without updating.
Thank you very much.

Round 2
Reviewer 2 Report
The revised version looks good to me!
Author Response
Dear Reviewer,
Thank you once again for your kind comments. When I checked the second review results, I could not find your comments. In order to make our draft better, we additionally checked and fixed the English grammar and spelling. If your second comment did not reach us for some reason, please send it again, and we will gladly revise it.
Thank you very much for taking the time out of your busy schedule.
Best regards,
Authors
